# Application of Additively Manufactured Pentamode Metamaterials in Sodium/Inconel 718 Heat Pipes

**DOI:** 10.3390/ma14113016

**Published:** 2021-06-02

**Authors:** Longfei Hu, Ketian Shi, Xiaoguang Luo, Jijun Yu, Bangcheng Ai, Chao Liu

**Affiliations:** Laboratory of Aero-Thermal Protection Technology for Aerospace Vehicles, China Academy of Aerospace Aerodynamics, Beijing 100074, China; shiketian@spacechina.com (K.S.); luoxiaoguang@spacechinaren.com (X.L.); yujijun@spacechinaren.com (J.Y.); aibangcheng@spacechina.com (B.A.); liuchao@spacechina.com (C.L.)

**Keywords:** heat pipes, pentamode metamaterials, sodium, Inconel 718, startup

## Abstract

In this study, pentamode metamaterials were proposed for thermal stress accommodation of alkali metal heat pipes. Sodium/Inconel 718 heat pipes with and without pentamode metamaterial reinforcement were designed and fabricated. Then, these heat pipes were characterized by startup tests and thermal response simulations. It was found that pentamode metamaterial reinforcement did not affect the startup properties of sodium/Inconel 718 heat pipes. At 650–950 °C heating, there was a successful startup of heat pipes with and without pentamode metamaterial reinforcement, displaying uniform temperature distributions. A further simulation indicated that pentamode metamaterials could accommodate thermal stresses in sodium/Inconel 718 heat pipes. With pentamode metamaterial reinforcement, stresses in the heat pipes decreased from 12.9–62.1 to 10.2–52.4 MPa. As a result, sodium/Inconel 718 heat pipes could be used more confidently. This work was instructive for the engineering application of alkali metal heat pipes.

## 1. Introduction

Heat pipes, combining the advantages of high conductivity and inherent safety, are attractive devices for thermal management of hot structures, such as large-scale heat exchangers, thermal protection of hypersonic vehicles, heat radiation of space reactors and so on [1,2,3,4]. For these applications, alkali metal heat pipes were feasible for source temperatures above 600 °C. Power throughputs of 15 kW/cm^2^ were possible for sodium heat pipes operated at 880 °C [5].

The feasibility of alkali metal heat pipes has been demonstrated by various researchers. Rosenfeld and his co-workers [6,7] evaluated sodium/Inconel 718 heat pipes for use as heat exchangers by 10 year life tests. For over 87,000 h at nearly 700 °C, no significant degradation of screen wicks or Inconel 718 walls were observed, providing strong evidence for chemical compatibility between sodium and Inconel 718 alloy. Camarda [8,9] fabricated sodium heat pipes for thermal protection of hypersonic vehicles. He verified that sodium heat pipes cooled leading edges sufficiently enough, allowing the use of nickel-based superalloys. Then, a leading edge-like heat pipe was fabricated, which reduced stagnation temperatures from 1926 to 900 °C [10,11]. Dussinger and his co-workers [12] fabricated alkali metal heat pipes as space radiators. Their experimental results implied that the capacity of potassium heat pipes was 5–15 kW, being larger than that of cesium heat pipes. However, severe temperature gradients were generated in the startup process of the above-mentioned alkali metal heat pipes, which might lead to thermal stresses and undesirable deformations.

The purpose of this study was to accommodate thermal stresses in alkali metal heat pipes. With thermal stress accommodation, undesirable deformations of heat pipes were reduced. Consequently, alkali metal heat pipes could be used more confidently in high temperature conditions. This study was meaningful for the development of advanced heat pipes, especially for that of advanced heat pipe spreaders [13,14,15]. 

In this study, additively manufactured pentamode metamaterials were proposed for thermal stress accommodation of alkali metal heat pipes. Sodium/Inconel 718 heat pipes with and without pentamode metamaterial reinforcement were designed and fabricated at the China Academy of Aerospace Aerodynamics (CAAA). Then, these heat pipes were characterized by startup tests and thermal response simulations. Finally, thermal stresses in fabricated sodium/Inconel 718 heat pipes were analyzed. 

## 2. Experimental

Figure 1 shows the illumination of a heat pipe operation [16]. In Figure 1, the startup process was as follows: Heat was absorbed by the evaporator of the heat pipe. The working fluid evaporated, leading to an internal pressure between the evaporator and the condenser. Then, vapor flew down to the condenser and gave up heat. Finally, the liquid working fluid returned to the evaporator via a capillary wick. During the startup processes, severe temperature gradients and thermal stresses were produced, which was detrimental to the engineering applications of the alkali metal heat pipes.

Concepts of pentamode metamaterials were proposed as early as 1995. In 2012, additive manufacturing enabled the realization of such light-weighted materials. Compared with traditional materials, the mechanical properties of pentamode metamaterials were linked to their macro-topology. With macro-topology adjustment, a negative thermal expansion and negative Poisson’s ratio were achieved [17,18,19]. Thus, thermal stresses and strains in structures might be accommodated by pentamode metamaterial reinforcement. 

### 2.1. Design of Heat Pipes With and Without Reinforcement

Heat pipes with dimensions of Φ25 × 500 mm were designed at our company. For the working fluid, 99.9% pure sodium was used. Inconel 718 and GH4169 (two kinds of nickel-based superalloys) were candidate materials of heat pipe walls and pentamode metamaterials. For additive manufacturing, only Inconel 718 powders were commercially obtained. Hence, Inconel 718 was used as the raw material of the heat pipes. The chemical compositions of Inconel 718 are listed in Table 1. Annular gaps with dimensions of 0.5 mm × 0.5 mm (width × height) were used as capillary wicks. 

Figure 2 shows the illumination of the designed heat pipes with and without pentamode metamaterial reinforcement. In Figure 2a, the designed heat pipes included an Inconel 718 wall, two endcaps, capillary wick (annular gaps), sodium, and pentamode metamaterial reinforcement. The Inconel 718 wall and endcaps were 1.5 mm in thickness. The capillary wick was 36 annular gaps, distributed evenly on the inner surface of the heat pipe wall. The topology of reinforcement was a lattice structure based on cubic unit cells with struts so as to form metamaterials. The cubic unit cells were 7 mm × 7 mm × 7 mm (length × width × height). The struts were 0.5 mm in diameter. Moreover, sodium/Inconel 718 heat pipes without pentamode metamaterial reinforcement were also designed, as seen in Figure 2b. In Figure 2b, the Inconel 718 wall, endcaps, and capillary wick were identical to that in Figure 2a. 

### 2.2. Fabrication of Heat Pipes with and without Reinforcement 

Sodium/Inconel 718 heat pipes were fabricated by the following process: (1) Inconel 718 walls with and without pentamode metamaterial reinforcement were additively manufactured by selective laser melting (Realizer SLM 125, Germany), and annular gaps were in situ formed on the inner surface of the heat pipe walls; (2) endcaps were machined using bulk Inconel 718 alloy; (3) Inconel 718 walls and endcaps were ultrasonically cleaned by acetone, ethanol, and de-ionized water, and then overnight dried in a 70° C oven; (4) an endcap was welded into the Inconel 718 wall, obtaining heat pipe shells for sodium charging; (5) sodium charging was performed via a vacuum/argon facility [20]; and (6) the other endcap was equipped on heat pipe shells, and then seal welded in a vacuum environment.

Four models of sodium/Inconel 718 heat pipes were fabricated, as seen in Table 2. In Table 2, HP1 and HP3 were with pentamode metamaterial reinforcement, while HP2 and HP4 were without pentamode metamaterial reinforcement. The amounts of 16 and 20 g sodium were charged into the heat pipes, with a corresponding charging ratio of 8 and 10%, respectively. The densities of HP1–HP4 were 2.02–2.04 g/cm^3^, being very close to each other. Pentamode metamaterial reinforcement displayed a negligible impact on model weight.

To ensure the amount of sodium charging, all the fabricated heat pipes (HP1–HP4) were weighted and inspected by X-ray. Then, these heat pipes were heated in a 600 °C oven, redistributing charged sodium uniformly in the wicks. Figure 3a shows an optical image of a fabricated heat pipe (HP1) and its cross section.

### 2.3. Characterization

#### 2.3.1. Startup Tests

Thermocouples (TCs) were used for the axial temperature checking of heat pipes. In this study, 6 K-type TCs were spot welded at regular intervals of 90 mm, as seen in Figure 3b. The diameter of these TCs were 250 μm. Before the startup tests, they were calibrated with an accuracy of ±7.5 °C.

Figure 4 shows a schematic diagram of the 30 kW calorifier and startup tests. In Figure 4, the calorifier was Φ300 mm × 300 mm. The porous Al_2_O_3_ container was 50 mm in thickness. A graphite oven was used as a radiator. During the startup tests, the graphite oven was heated to temperatures of 650–950 °C. Then, the evaporator of the heat pipe (250 mm long from one endcap) was inserted into the oven to check the thermal response of all the TCs.

In this study, sodium/Inconel 718 heat pipes were tested as follows: (1) heated at a temperature of 650 °C; (2) heated at a temperature of 750 °C; (3) heated at a temperature of 850 °C; and (4) heated at a temperature of 950 °C.

#### 2.3.2. Thermal Response Simulation

An engineering method, named as the flat-front model, was used to simulate the thermal response of sodium/Inconel 718 heat pipes. In this model, all the nodes were divided according to heating characteristics, and the flat-front position was determined by the local energy conservation theory. Details about the flat-front model and its simulation can be found elsewhere [21].

Based on the flat-front model, an in-house software, named as thermal analysis heat pipe (TAHP), was developed for the thermal response simulation [22]. During the thermal response simulation, both thermal radiation and convective cooling were considered. Emissivity (ε) of thermal radiation was assumed to be 0.9 (emissivity of Al_2_O_3_ scale formed on the heat pipe surfaces). Film coefficient (h) of convective cooling was assumed to be 20 W·m^−2^·°C^−1^ (film coefficient of Inconel 718).

The thermal response of the sodium/Inconel 718 heat pipes indicated that the simulated results agreed well with the startup results, identifying the validity of the flat-front model. Then, an endcap surface was fixed supported, calculating the thermal stresses and deformation of sodium/Inconel 718 heat pipes. Finally, the stress concentration in the heat pipes was analyzed, and the effects of pentamode metamaterial reinforcement were discussed.

## 3. Results and Discussion

### 3.1. Startup Results

HP1 and HP2 were heated at 850 °C, and their optical images are shown in Figure 5. In Figure 5a, we can see that HP1 displayed a bright evaporator, bright adiabatic, and dark condenser, indicating the existence of asymmetric heating. In contrast, HP2 displayed a bright evaporator, bright adiabatic, and bright condenser, as seen in Figure 5b. Therefore, HP2 displayed a more uniform temperature distribution. This was most probably caused by the startup of HP2, transferring heat efficiently from the evaporator to the adiabatic and the condenser [23].

Figure 6 shows the startup results of HP1 and HP2 at 850 °C heating. In Figure 6a, HP1 was heated for 600 s. In the initial 0–150 s, the temperatures of TC1–TC4 increased greatly to 700–750 °C. The temperatures of TC5 and TC6 were about 600 and 22 °C, being significantly lower than that of TC1–TC4. In the following 150–600s heating, TC1–TC6 displayed temperature differences (Δ*T*) of 450–800 °C, indicating that startup failures occurred for HP1. In Figure 6b, the successful startup of HP2 is shown. In the initial 0–130 s, all the TC temperatures increased greatly to an identical value of ~600 °C. Then, these temperatures increased gradually to ~830 °C in the following tests. For sodium charging of 8%, it appeared that sodium/Inconel 718 heat pipes without pentamode metamaterial reinforcement showed better startup properties.

In Figure 6a, a noticeable bump occurred in the TC1 measurement. A possible explanation was that excessive gas remained in HP1 (as seen in Section 3.3.1). In the initial 0–80 s of heating, the heat was absorbed by sodium evaporation. An internal pressure formed between the evaporator and the condenser. However, this pressure was lower than the remaining gas pressure. Sodium vapor could not flow from the evaporator to the condenser. The temperatures of TC1–TC4 increased greatly, especially for that of HP1. In the following 80–600 s of heating, the internal pressure was higher than the remaining gas pressure. Sodium vapor flew from the evaporator to the condenser. The temperatures of TC1–TC4 became close to each other. Therefore, a noticeable bump occurred in the TC1 measurement.

Figure 7 shows the startup results of HP3 and HP4 at 850 °C heating. In Figure 7, the successful startup of both HP3 and HP4 is shown. After 200–300 s of heating, the two heat pipes displayed uniform temperatures of 800–830 °C. It was noted that the startup time of HP3 was longer than that of HP4. This was probably caused by pentamode metamaterial reinforcement, which might increase the startup temperature and elongate the startup time of sodium/Inconel 718 heat pipes [23,24]. Details of this phenomenon are discussed in Section 3.3.1.

HP2, HP3, and HP4 were heated at 650, 750, 850, and 950 °C, respectively. Their startup properties are shown in Figure 8. At 650–950 °C heating, the temperature difference (Δ*T*) of HP2–HP4 was kept at 30–50 °C, but power throughputs increased from 1.6 to 4.8 kW. As HP2–HP4 displayed similar startup properties, it was concluded that pentamode metamaterial reinforcement had negligible impact on the startup properties of heat pipes.

Figure 9 shows optical images of tested HP1–HP4. In Figure 9, we can see that all tested heat pipes still kept their integrity. Structure failures or sodium leakage were not found. At 650–950 °C heating, thermal stresses in sodium/Inconel 718 heat pipes would be lower than the strength of the Inconel 718 alloy.

### 3.2. Simulation Results

HP3 and HP4 were selected for the thermal response simulation because of their identical sodium charging ratio. According to the flat-front model and heating characteristics, hex-dominant mesh was applied for HP3 and HP4. The maximum grid size was 5 × 10^−3^ m. The parameters of the thermal response are listed in Table 3. Figure 10 shows thermal stresses (Mises) in sodium/Inconel 718 heat pipes at 850 °C heating. For the HP3 startup, an area of stress concentration was produced in the heat pipe wall. Then, this area moved directly to the endcap around the condenser. Maximum stresses of 29.2 MPa appeared in the location of TC6. For the HP4 startup, an area of stress concentration was also produced and then moved to the endcap. The maximum stresses located in TC6 were 40.7 MPa, being higher than that in HP3. Thus, it was concluded that pentamode metamaterials could accommodate thermal stresses in sodium/Inconel 718 heat pipes. A further investigation indicated that this accommodation was related to pentamode metamaterial reinforcement (as seen in the zoomed-in images in Figure 10), which are discussed in Section 3.3.2.

Figure 11 shows Mises stresses located in TC6 of the sodium/Inconel 718 heat pipes. Mises stresses were unidirectional. At 650 °C heating, Mises stresses in HP3 were 10.2 MPa. At 750, 850, and 950 °C heating, Mises stresses in HP3 increased to 16.0, 29.2, and 52.4 MPa, respectively. On the other hand, Mises stresses in HP4 were 12.9–62.1 MPa, being about 30% higher than that in HP3. With stress accommodation, sodium/Inconel 718 heat pipes with pentamode metamaterial reinforcement (HP3) could be used more confidently.

Figure 12 shows the axial deformation of HP3 and HP4 at 650–950 °C heating. The direction of the axial deformation was from the evaporator to the condenser (from TC1 to TC6). In Figure 12, we can see that pentamode metamaterial reinforcement did not affect axial deformation of sodium/Inconel 718 heat pipes. At 650 °C heating, the axial deformation of HP3 was 5.07 mm. As the heating temperature rose, the axial deformation of HP3 increased linearly to 8.01 mm. Moreover, the axial deformation of HP4 was almost the same as that of HP3. This was beneficial for engineering applications of sodium/Inconel 718 heat pipes with pentamode metamaterial reinforcement. 

### 3.3. Discussion

#### 3.3.1. Startup Temperatures

According to previous studies [23,25], the saturation pressure of sodium heat pipes was calculated by the following equation:(1)$$\mathrm{lg}P_{s}=A-\frac{B}{T}$$
where $P_{s}$ is the saturation pressure of sodium (×10^5^ Pa); T is the operating temperature of the sodium heat pipes (K); and *A* and *B* are constant, equal to 4.544579 and 5242.1, respectively. The startup temperatures (*T**) of the sodium heat pipes were determined by the following equation [25]: (2)$$T*=\frac{\sqrt{2}\pi d^{2}K_{n}P_{s}D_{v}}{1.051\kappa }$$
where *d* is the effective molecular diameter of sodium, equal to 3.567 × 10^−10^ m; *K_n_* is the Knudsen number, equal to 0.01; *D_v_* is the vapor passage diameter or equivalent vapor passage diameter, m; and *κ* is the Stefan–Boltzmann constant, equal to 5.67 × 10^−8^ W·m^−2^·K^−4^. The calculated startup temperatures of HP2–HP4 were 417, 426, and 417 °C, respectively. These temperatures were lower than the heating temperatures in Section 2.3.1. As a result, there was a successful startup of sodium/Inconel 718 heat pipes in the startup tests.

During the process of startup tests, heat was absorbed by evaporation of liquid sodium. This evaporation resulted in an internal pressure between the evaporator and the condenser, which caused the vapor to flow down to the condenser and give up heat. However, pentamode metamaterial reinforcement could slow down the vapor flow from the evaporator to the condenser. Consequently, the startup temperature of HP3 became higher than that of HP2 and HP4, and the startup time was elongated. A further examination implied that excessive gas remained in the shell of HP1, leading to startup failures of the sodium/Inconel 718 heat pipe.

#### 3.3.2. Effects of Pentamode Metamaterial Reinforcement

Table 4 lists the high temperature strength of Inconel 718 alloy [26]. At heating temperatures of 650–950 °C, the strength of the Inconel 718 alloy was 109–970 MPa. These values are greater than the thermal stresses in Section 3.2. Therefore, all tested heat pipes still kept their integrity.

According to Equation (1), we know that sodium pressure (*P_s_*) in this study was about 4.6 × 10^4^ Pa, being lower than atmosphere pressure (1.0 × 10^5^ Pa). Consequently, a pressure of 5 MPa was put on the outside walls of HP3 and HP4, illuminating the effects of pentamode metamaterial reinforcement. Elastic strains of HP3 and HP4 were transiently simulated in the in-house software of thermal analysis heat pipe (TAHP), as seen in Figure 13. In Figure 13, the elastic strains of HP4 were 2.37 × 10^−4^ mm/mm. With pentamode metamaterial reinforcement, elastic strains of HP3 reduced to 1.99 × 10^−4^ mm/mm. Moreover, uniform strains were produced in the reinforcement. With elastic strain reduction, stresses in the heat pipes were accommodated.

The effects of pentamode metamaterial reinforcement were probably as follows: During startup tests, heat was transferred from the evaporator to the condenser. Stress concentration and elastic strains were produced in an area of the heat pipe walls. This area moved from the evaporator to the condenser. With pentamode metamaterial reinforcement, elastic strains occurred not only in the heat pipe walls but also in the reinforcement, as seen in the zoomed-in images in Figure 10. Because Poisson’s ratio of pentamode metamaterials was negative, strain areas in the reinforcement were larger than that in the heat pipe walls [19,27,28]. Therefore, stresses in the heat pipes were accommodated. On the other hand, strain variation did not affect the deformation of heat pipes, as HP3 and HP4 showed identical axial deformation.

#### 3.3.3. Future Applications

According to above analysis, it was demonstrated that pentamode metamaterial reinforcement could accommodate thermal stresses in sodium/Inconel 718 heat pipes; hence, this was a solution to the thermal-structure problems of alloys. Other potential applications (using different topological structures and different heat pipes) included: limiting thermal distortions of hot structures [28], cooling of space reactors, and thermal management of hypersonic vehicles. Recently, heat pipes with pentamode reinforcement were designed for space reactor cooling.

## 4. Conclusions 

In this study, pentamode metamaterials were proposed for thermal stress accommodation of alkali metal heat pipes. Sodium/Inconel 718 heat pipes with and without pentamode metamaterial reinforcement were designed and fabricated. These heat pipes were characterized by startup tests and thermal response simulations. Based on the experimental and theoretical results, the following conclusions were drawn: (1)Pentamode metamaterial reinforcement displayed a negligible impact on model weight. The densities of HP1–HP4 were 2.02–2.04 g/cm^3^, being very close to each other.(2)The startup temperatures of HP2–HP4 were 417, 426, and 417 °C, being lower than the heating temperatures. During the startup tests, there was a successful startup of these heat pipes.(3)Excessive gas remained in the shell of HP1, leading to startup failures of sodium/Inconel 718 heat pipes.(4)Pentamode metamaterials could accommodate thermal stresses in sodium/Inconel 718 heat pipes. With pentamode metamaterial reinforcement, thermal stresses in the heat pipes reduced from 12.9–62.1 to 10.2–52.4 MPa.(5)With thermal stress accommodation, sodium/Inconel 718 heat pipes could be used more confidently.

## Figures and Tables

**Figure 1 materials-14-03016-f001:**
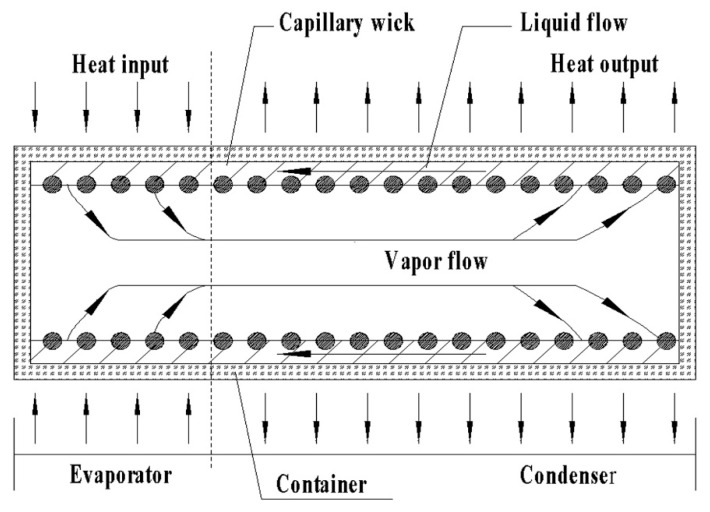
Illumination of a heat pipe operation, reference [16].

**Figure 2 materials-14-03016-f002:**
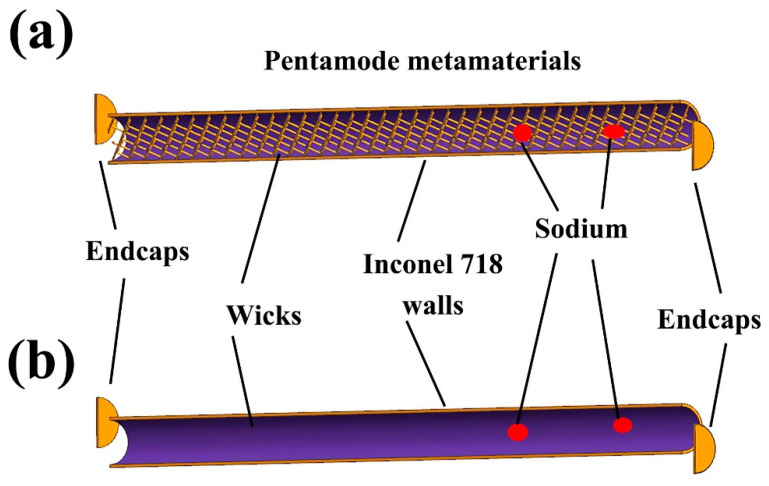
Illumination of designed heat pipes: (**a**) with pentamode metamaterial reinforcement, (**b**) without pentamode metamaterial reinforcement.

**Figure 3 materials-14-03016-f003:**
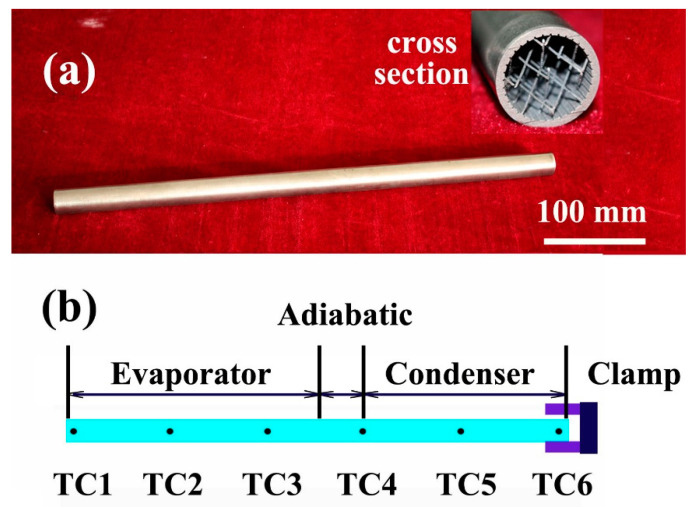
An optical image of a fabricated heat pipe (**a**) and the thermocouples’ location (**b**).

**Figure 4 materials-14-03016-f004:**
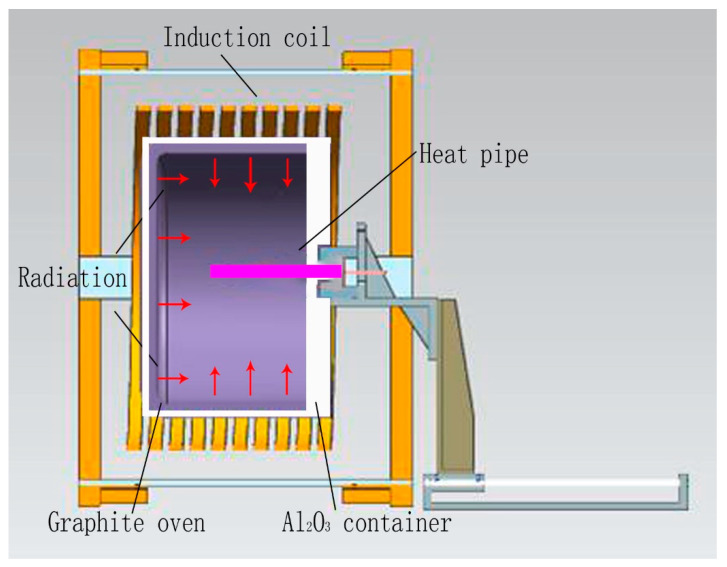
Schematic diagram of 30 kW calorifier and startup tests.

**Figure 5 materials-14-03016-f005:**
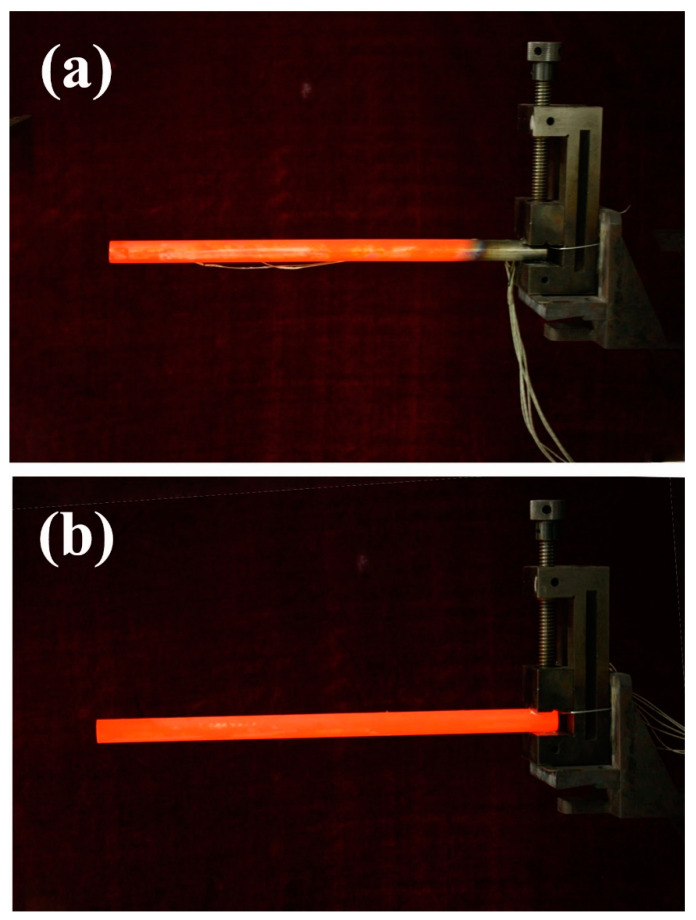
Optical images of heat pipes at 850 °C heating: (**a**) HP1, (**b**) HP2.

**Figure 6 materials-14-03016-f006:**
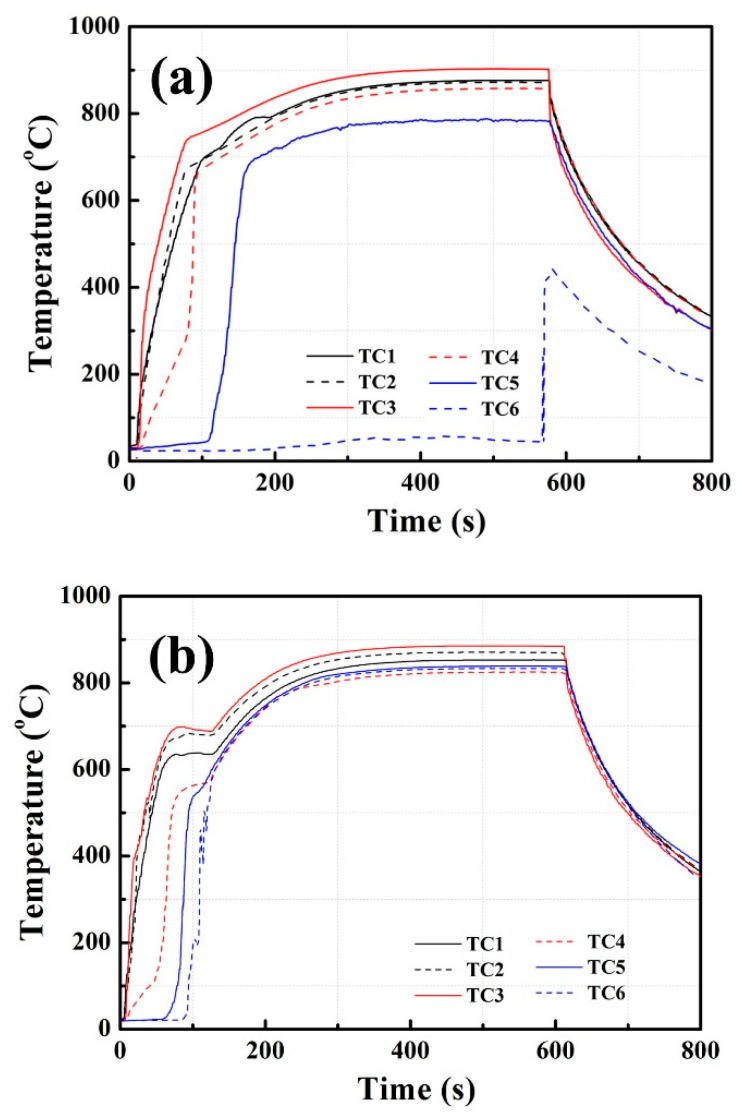
Startup results of heat pipes at 850 °C heating: (**a**) HP1, (**b**) HP2.

**Figure 7 materials-14-03016-f007:**
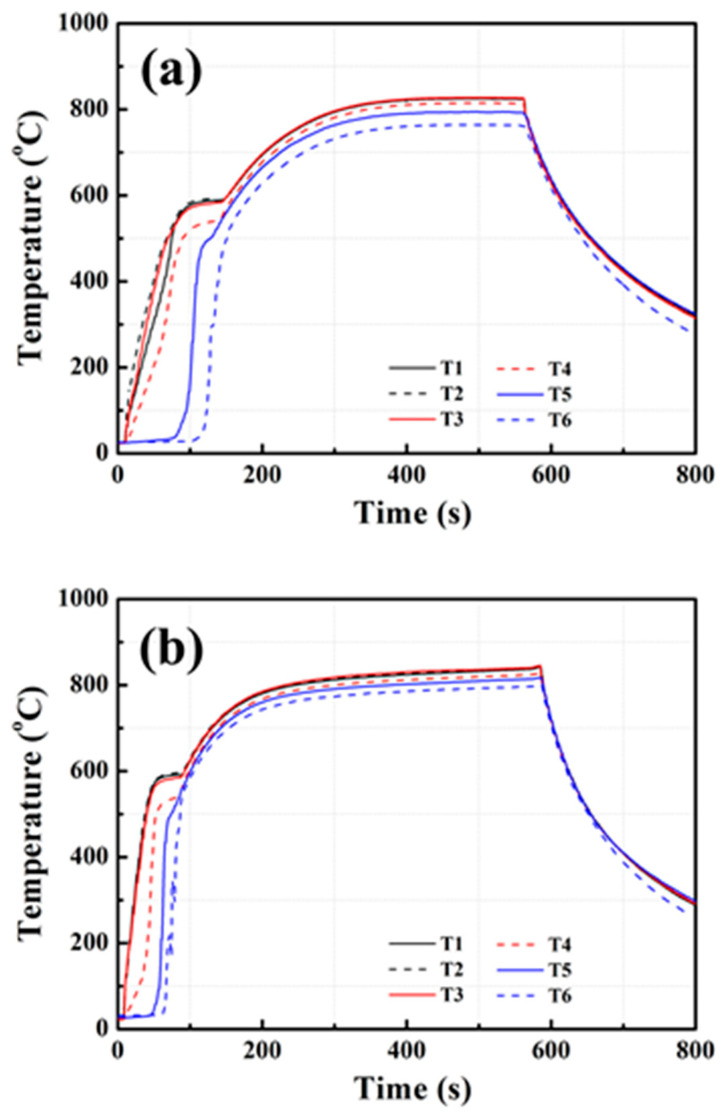
Startup results of heat pipes at 850 °C heating: (**a**) HP3, (**b**) HP4.

**Figure 8 materials-14-03016-f008:**
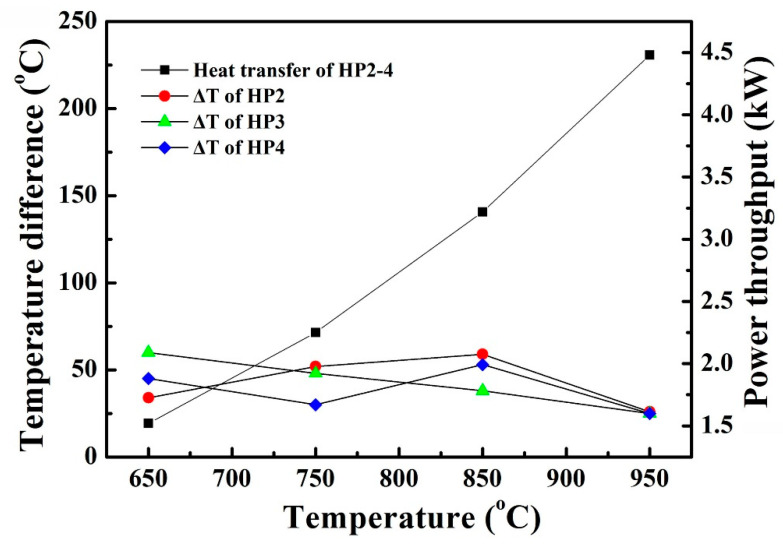
Startup properties of HP2–HP4 at 650–950 °C heating.

**Figure 9 materials-14-03016-f009:**
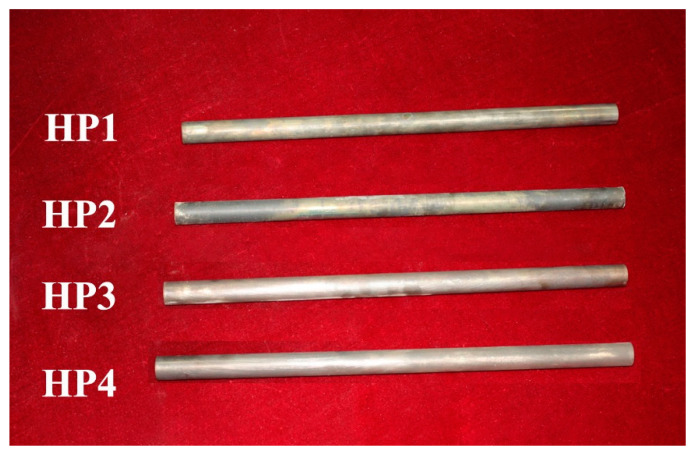
Optical images of tested HP1–HP4.

**Figure 10 materials-14-03016-f010:**
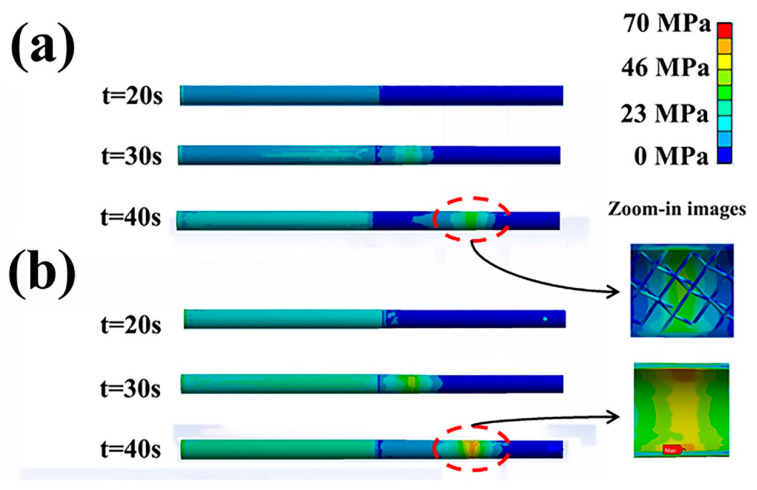
Thermal stresses (Mises) in sodium/Inconel 718 heat pipes at 850 °C heating: (**a**) HP3, (**b**) HP4.

**Figure 11 materials-14-03016-f011:**
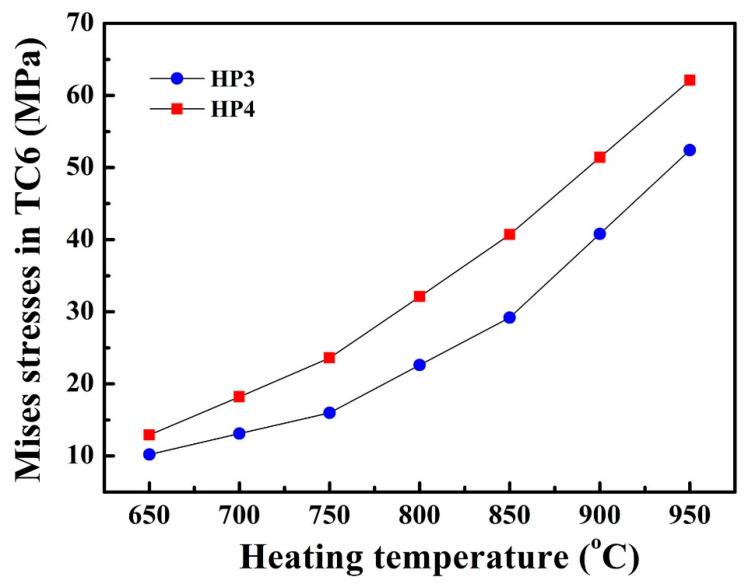
Mises stresses located in TC6 of sodium/Inconel 718 heat pipes.

**Figure 12 materials-14-03016-f012:**
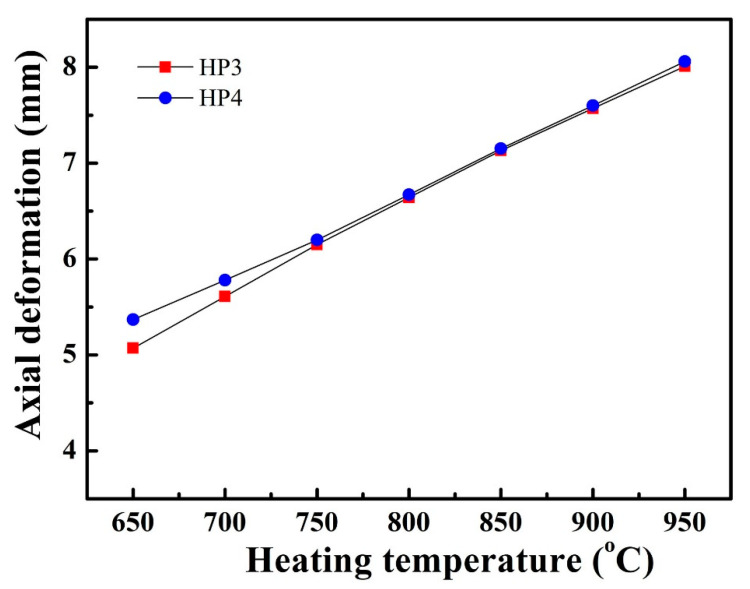
Axial deformation of HP3 and HP4 at 650–950 °C heating.

**Figure 13 materials-14-03016-f013:**
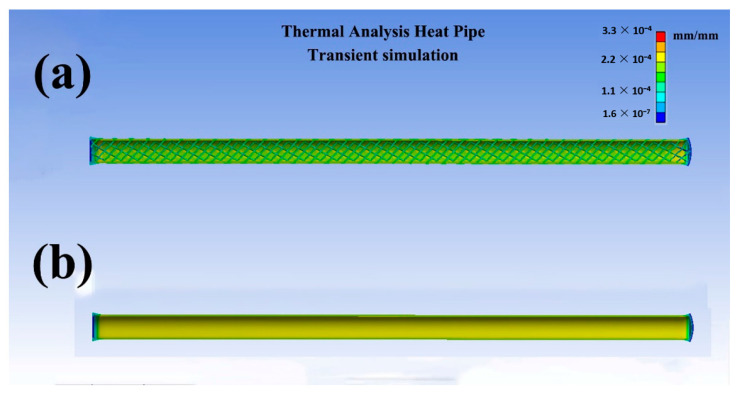
Elastic strains under a pressure of 5 MPa: (**a**) HP3, (**b**) HP4.

**Table 1 materials-14-03016-t001:** Chemical compositions of Inconel 718.

**Element**	Ni	Fe	Cr	Cu	Mo	Nb
**Weight Percent (%)**	50–55	remainder	17–21	≤0.3	2.8–3.3	4.75–5.5
**Element**	C	Mn	P	S	Si	Ti
**Weight Percent (%)**	≤0.08	≤0.35	≤0.015	≤0.015	≤0.35	0.65–1.15
**Element**	Al	Co	B			
**Weight Percent (%)**	0.20–0.80	≤1.00	≤0.006			

**Table 2 materials-14-03016-t002:** Fabricated heat pipes with and without pentamode metamaterial reinforcement.

Models	Model Weight (g)	Charged Sodium (g)	Charging Ratio (%)	Density (g/cm^3^)	Pentamode Metamaterials
HP1	500.3	16	8	2.04	With
HP2	495.0	16	8	2.02	Without
HP3	498.2	20	10	2.03	With
HP4	497.2	20	10	2.03	Without

**Table 3 materials-14-03016-t003:** Parameters of thermal response simulation.

**Model**	HP3/HP4	**Heating time**	600 s
**Evaporator**	250 mm	**Emissivity**	0.9
**Adiabatic**	50 mm	**Film coefficient**	20 W/m^−2^ °C^−1^
**Condenser**	200 mm	**Grid size**	≤5 × 10^−3^ m
**Heating temperatures**	650 °C/750 °C/ 850 °C/950 °C	**Mesh type**	Hex dominant

**Table 4 materials-14-03016-t004:** High temperature strength of Inconel 718 alloy [26].

Temperature (°C)	Strength (MPa)	Temperature (°C)	Strength (MPa)
20	1200	850	299
600	1040	900	180
650	970	950	109
750	710		

## Data Availability

The data presented in this study are available on request from the corresponding author.

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
