# Peer review of "Application of Additively Manufactured Pentamode Metamaterials in Sodium/Inconel 718 Heat Pipes"

_materials, 2021, doi:10.3390/ma14113016_

Round 1

Reviewer 1 Report

Dear Authors,

I have read your paper "Application of Additively Manufactured Pentamode Metamaterials in Sodium/Inconel 718 Heat Pipes" carefully. 

The paper is easy to read.

Methods are properly described, so that other research groups may reproduce them.

The paper is interesting. However, it requires few corrections.

  1. Please, add a paragraph with the purpose of the work.
  2. Please, add information about the chemical composition of the Inconel 718. 
  3. Please, add information about GH4169 in the text. 

 The paper can be accepted for publication after minor improvements.

Author Response

Replies to Reviewer  #1

I have read your paper "Application of Additively Manufactured Pentamode Metamaterials in Sodium/Inconel718 Heat Pipes" carefully. The paper is easy to read. Methods are properly described so that other research groups may reproduce them. This paper is interesting. However, it requires few corrections.

Comment 1.  Please, add a paragraph with the purpose of the work.

Reply:  Correct

In the revised manuscript, the purpose of this study was detailed in Page 2, Paragraph 1 (line 45-51).

The purposes of this study were accommodating thermal stresses in alkali-metal heat pipes. With thermal stress accommodation, undesirable deformations of heat pipes were reduced. Consequently, alkali-metal heat pipes could be used more confidently in high temperature conditions. This study was meaningful for development of advanced heat pipes, especially for that of advanced heat pipe spreaders [13-15].  

Comment 2.  Please add information about the chemical compositions of the Inconel 718.

Reply:  Correct.

In the revised manuscript, chemical compositions of Inconel 718 was added in Table 1.

Table 1 chemical compositions of Inconel 718

Element

Ni

Fe

Cr

Cu

Mo

Nb

Weight percent (%)

50-55

remainder

17-21

≤0.3

2.8-3.3

4.75-5.5

Element

C

Mn

P

S

Si

Ti

Weight percent (%)

≤0.08

≤0.35

≤0.015

≤0.015

≤0.35

0.65-1.15

Element

Al

Co

B

Weight percent (%)

0.20-0.80

≤1.00

≤0.006

Comment 3.  Please, add information of GH4169 in the text.

Reply:  Correct

Information of GH4169 were added in Page2, last Paragraph.

Dimension of Ф25×500 mm heat pipes were designed at our company. 99.9% pure sodium was used as the working fluid. Inconel 718 and GH4169 (two kinds of nickel-based superalloys) were candidate materials of heat pipe walls and pentamode metamaterials. For additively manufacture, only Inconel 718 powders was commercial obtained. Hence, Inconel 718 was used as raw materials of heat pipes. Chemical compositions of Inconel 718 were listed in Table 1. Annular gaps with dimension of 0.5 mm×0.5 mm (width×height) were used as capillary wicks. 

Reviewer 2 Report

Good effort. This paper has the potential to be improved but will require some work before it can be published. 

1) Software, model and mesh should be adequately emphasized

2) Language should be polished

3) Diagrams need to be significantly improved, apply scales where required. 

Author Response

Replies to Reviewer  #2

Comment 1. In Page 4, can Fig. 3a come with a scale bar? That would be clear, also would appreciate if there could be some form of labels to clearly indicate what these sections are?

In Page 4 Fig. 3, is TC6 or TC1 being clamped? Can indicate on Fig. 3b?

Reply:  Correct

In revised manuscript, a scale bar was added in Fig. 3a. And labels were involved in the revised Fig.3, indicating evaporator, adiabatic, and condenser section of heat pipes.

In the revised manuscript, clamped TC6 was indicated in Fig.3b, seen in Page 4.

Fig. 3

Comment 2.  In Page 5, 2.3.2 thermal response simulation, can the author indicate what is the soft utilized here? Is it an in-house software? Ansys or Comsol? Thanks!

Reply:  Correct

In the revised manuscript, it was indicated that an in-house software was utilized for the thermal response simulation of heat pipes, seen in Page 5, Paragraph 2, line 135-140.

Based on flat-front model, an in-house software, named as thermal analysis heat pipe (TAHP), was developed for thermal response simulation [22]. During thermal response simulation … …

Comment 3.  In Page 5, Paragraph 2, line 40, "Emissivity (ε) of thermal radiation was chosen as 0.9." can be revised as "Emissivity (ε) of thermal radiation was assumed to be 0.9.",  good to highlight source also.

Reply:  Correct

In the revised manuscript, the description of emissivity was revised, its source was highlighted.

Emissivity (ε) of thermal radiation was assumed to be 0.9 (emissivity of Al2O3 scale formed on the heat pipe surfaces).

Comment 4.  In Page 6, Paragraph 1, line 168, "it was seemed that sodium/Inconel 718 heat pipes… ..." Rephrase please, consider  "it appeared that".

Reply:  Correct

In the revised manuscript, the description was rephrased as following:  

For sodium charging of 8%, it appeared that sodium/Inconel 718 heat pipes without pentamode metamaterials reinforcement showed better startup properties.

Comment 5.  In Page 7  Fig. 6a, TC1 with a noticeable bump in its measurement. Any explanations

Reply:  Correct

In revised manuscript, a new paragraph was added in Page 6 line 171-174, explaining the notice bump of TC1.

In Fig. 6a, a noticeable bump occurred in TC1 measurement. A possible explanation was that excessive gas remained in HP1 (seen in 3.3.1). In the initial 0-80s heating, heat was absorbed by sodium evaporation. An internal pressure formed between evaporator and condenser. However, this pressure was lower than the remained gas pressure. Sodium vapor could not flow from evaporator to condenser. Temperatures of TC1-4 increased greatly, especially for that of HP1. In the following 80-600s heating, the internal pressure was higher than the remained gas pressure. Sodium vapor flew from evaporator to condenser. Temperatures of TC1-4 became close to each other. Therefore, a noticeable bump occurred in TC1 measurement.

Comment 6.  In Page 9 Fig.9, HP1, HP2, HP3, and HP4 are not clearly indicated.

Reply:  Correct

In the revised manuscript, labels of HP1, HP2, HP3, and HP4 were removed to the left side of Fig. 9, indicated clearly to the readers, seen in Page 9 Fig.9.

Fig. 9

Comment 7.  In Page 9 section 3.2, authors to consider, explaining thermal response simulation in details:

(1) What kind of mesh they are applying and why?

(2) In Fig. 10, are these stresses the ones observed in Fig. 11? Maybe good to clearly indicated area of study for thermal stresses.

(3) To correlate to HP1, HP2, HP3, and HP4, better, also consider labeling your simulation.

Reply:  Correct

(1) The kind of mesh and why were detailed in Page 9, Paragraph 1, line 204-206. 

HP3 and HP4 were selected for thermal response simulation because of their identical sodium charging ratio. According to flat-front model and heating characteristics, hex dominant mesh was applied for HP3 and HP4 simulation. The maximum grid size was 5×10-3 m… …

(2) In Fig. 10, these stresses were the ones observed in Fig. 11. To clearly indicate area of study, maximum stress located in TC were descripted in Page 9, Paragraph 1, line 206-210, seen below:

Fig. 10 shows thermal stresses (Mises) in sodium/Inconel 718 heat pipes at 850oC heating. For HP3 startup, an area of stress concentration was produced in the heat pipe wall. Then this area moved directly to the endcap around condenser. Maximum stresses of 29.2 MPa appeared in the location of TC6. For HP4 startup, an area of stress concentration was also produced and then moved to the endcap. The maximum stresses located in TC6 were 40.7 MPa, being higher than that in HP3.

(3) To correlate to HP1, HP2, HP3, and HP4, better, labeling of simulation was displayed in Table .3.

Table 3 Parameters used for thermal response simulation

Model

HP 3/ HP4

Heating time

600 s

Evaporator

250 mm

Emissivity

0.9

Adiabatic

50 mm

Film coefficient

20 W/m-2 oC-1

Condenser

200 mm

Grid size

≤5×10-3 m

Heating

temperatures

650oC/750oC/

850oC/950oC

Mesh type

Hex dominant

Comment 8.  In Page10 Fig. 11, it is good to show the location of HP3 and HP4 here. Indicate clearly on diagram.

Reply:  Correct

In the revised manuscript, stresses were located in TC6 of HP3 and HP4, seen Fig. 11. And the TC6 location was indicated clearly in Fig. 3.

Fig. 11

Comment 9.  In Page 12 Fig. 13, can the authors state the name of the software used and the type of simulation.  

Reply:  Correct

In the revised manuscript, software name of "Thermal Analysis Heat pipe" and type of "transient simulation" were stated in Fig. 13. This statement was also described in Page 11, last paragraph, line 260-261:

To illuminate effects of pentamode metamaterials reinforcement, a pressure of 5 MPa was put on the outside walls of HP3 and HP4, and their elastic strains were transient simulated in Thermal Analysis Heat Pipe (TAHP), seen in Fig. 13.

Fig. 13

Reviewer 3 Report

This article presents a novel design of heat pipe reinforced with a pentamode metamaterials infill. Thermal and mechanical properties are characterized in the application settings. The results presented support the conclusion that the pentamode metamaterial infill can alleviate the unfavored stress distribution in the heat pipe during its operation.

A few questions and suggestions listed as the following:

  1. The experimental design stated that HP1 and HP2 are both manufactured with pentamode metamaterial infill, but the results suggested that HP1 and HP3 are the two with infill (line 201-203). Please clarify in Table 1 and double-check throughout the manuscript.  
  2. Please specify the direction of the stress and deformation in Figures 11 and 12, respectively. Also updated the corresponding sections in the results and discussion.
  3. What exactly does the pentamode infill do to the structure? Would the authors provide a zoom-in image of the simulation and corresponding analysis focusing on the pentamode materials, in addition to the one currently only showing the entire tube?
  4. The authors presented a relatively unclear discussion about the effect of pentamode infill on the mechanical behavior of the heat pipe. Which properties of the pentamode metamaterials (negative Poisson's ratio, or negative thermal expansion? or combination of them) cause the reduction of stress?
  5. In section 3.3.2 the authors ran the simulation with applied external pressure. During the operation of the heat pipe, the pipe should go under internal pressure. Could the authors discuss and bridge this discrepancy?
  6. A question just for curiosity, normally the heat pipe will be manufactured through an extruding process? Would it be possible to introduce the pentamode metamaterial infill after the tube is extruded?
  7. formating suggestions: ---Figure 2: caption of (b) is missing. ---Figure 3b: could the authors mark the end of heating?  ---Figure 6: a, b are missing; please move the legend in figure 6a to reveal the entire T6 curve.
  8. Minor English language suggestion: please check the use of articles and tense throughout the manuscript. 

Author Response

Replies to Reviewer  #3

This article presents a novel design of heat pipe reinforced with a pentamode metamaterials infill. Thermal and mechanical properties are characterization in the application setting. The results presented support the conclusion that the pentamode metamaterial infill can alleviate the unfavoured stress distribution in the heat pipe during its operation. 

A few questions and suggestions listed as the following:

Comment 1.  The experiment design stated that HP1 and HP2 are both manufactured with pentamode metamaterial infill, but the results suggested that HP1 and HP3 are the two with infill (line 201-203). Please clarify in Table 1 and double-check throughout the manuscript.

Reply:  Correct

In the revised manuscript, HP1, HP2, HP3, and HP4 were clarified in Table 2 Heat pipes with and without pentamode metametarials reinforcement). Then the manuscript was double-check. Statement in experiment design was rewritten as: 

In Table 2, HP1 and HP3 were with pentamode metamaterials reinforcement, while HP2 and HP4 were without pentamode metamaterials reinforcement.

Table 2 Fabricated heat pipes with and without pentamode metametarials reinforcement

Models

Model

weight (g)

Charged

sodium (g)

Charging

ratio (%)

Density

(g/cm3)

Pentamode

Metamaterials

HP1

500.3

16

8

2.04

With

HP2

495.0

16

8

2.02

Without

HP3

498.2

20

10

2.03

With

HP4

497.2

20

10

2.03

Without

Comment 2.  Please specify the direction of the stress and deformation in Fig. 11 and Fig. 12, respectively. Also updated the corresponding sections in the results and discussion.

Reply:  Correct

In the revised manuscript, stress and deformation in Fig. 11 and Fig. 12 were specified. Then the corresponding section in the results and discussion was updated, seen in Page 9-10, line 214-228, Fig. 11, and Fig. 12.

Fig. 11 shows Mises stresses located in TC6 of sodium/Inconel 718 heat pipes. Mises stresses were unidirectional. At 650oC heating, Mises stresses in HP3 were 10.2 MPa. At 750oC, 850oC, and 950oC heating, Mises stresses in HP3 increased to 16.0 MPa, 29.2 MPa, and 52.4 MPa, respectively. On the other hand, Mises stresses in HP4 were 12.9-62.1 MPa, being about 30% higher than that in HP3. With stress accommodation, sodium/Inconel 718 heat pipes with pentamode metamaterials reinforcement (HP3) could be used more confidently.

Fig. 12 shows axial deformation of HP3 and HP4 at 650-950oC heating. The direction of axial deformation was from evaporator to condenser (from TC1 to TC6). In Fig. 12, we could see that pentamode metamaterials reinforcement did not affect axial deformation of sodium/Inconel 718 heat pipes. At 650oC heating, axial deformation of HP3 was 5.07 mm. As heating temperature rose, the axial deformation of HP3 increased linearly to 8.01 mm. Moreover, axial deformation of HP4 was almost the same to that of HP3. This was beneficial for engineering applications of sodium/Inconel 718 heat pipes with pentamode metamaterials reinforcement.

Fig. 11

Fig. 12

Comment 3.  What exactly does the pentamode infill do to the structure? Would the authors provide a zoom-in image of the simulation and corresponding analysis focusing on the pentamode metamaterials, in addition to the one currently only showing the entire tube?

Reply:  Correct

In the revised manuscript, a zoom-in image of the simulation was provided in Fig. 10, explaining what exactly does pentamode metamaterials infill do. Then corresponding analysis focusing on the pentamode metamaterials were added, seen below:

Analysis added in Page 9, first Paragraph, line 211.  

A further investigation indicated that this accommodation was related to pentamode metamaterials reinforcement (seen zoom-in images in Fig. 10), which would be discussed in section 3.3.2.

Analysis focusing on the pentamode metamaterials was added in Page 11, section 3.3.1, line 267.

With pentamode metamaterials reinforcement, elastic strains occurred not only in the heat pipe walls, but also in the reinforcement, seen zoom-in images in Fig. 10. Because Poisson´s ratio of pentamode metamaterials was negative, strain areas in the reinforcement were larger than that in the heat pipe walls [19, 27, 28]. Therefore, stresses in the heat pipes were accommodated.

Fig. 10 Thermal stresses (Mises) in sodium/Inconel 718 heat pipes at 850oC heating.

Comment 4.  The authors presented a relatively unclearly discussion about the effect of pentamode infill on the mechanical behavior of the heat pipe. Which properties of the pentamode metamaterials (negative Poission̕s ratio, or negative thermal expansion? Or combination of them) cause the reduction of stress?

Reply:  Correct

A new paragraph was added in Page 11, section 3.3.1, line 267, indicating that negative Poission̕s ratio cause the reduction of stress.

Effects of pentamode metamaterials reinforcement were properly as following. During startup tests, heat was transferred from evaporator to condenser. Stress concentration and elastic strains were produced in an area of heat pipe walls. And this area moved from evaporator to condenser. With pentamode metamaterials reinforcement, elastic strains occurred not only in the heat pipe walls, but also in the reinforcement, seen zoom-in images in Fig. 10. Because Poisson´s ratio of pentamode metamaterials was negative, strain areas in the reinforcement were bigger than that in the heat pipe walls [19, 27, 28]. Therefore, stresses in the heat pipes were accommodated.

Comment 5.  In section 3.3.2, the authors ran the simulation with applied external pressure. During the operation of heat pipe, the pipe should go under internal pressure. Could the authors discuss and bridge this discrepancy?

Reply:  Correct

In the revised manuscript, pressure discrepancy was discussed in section 3.3.2, Page 11, line 260-263.

According to equation (1), we could know that sodium pressure (Ps) in this study was about 4.6×104 Pa, being lower than atmosphere pressure (1.0×105 Pa). Consequently, a pressure of 5 MPa was put on the outside walls of HP3 and HP4, illuminating effects of pentamode metamaterials reinforcement.

Comment 6.  A question just for curiosity, normal the heat pipes will be manufactured through an extruding process? Would it be possible to introduce the pentamode metamaterial infill after the tube is extruded?

Reply:  Correct

In the revised manuscript, pentamode metamaterial was induced, seen in Page 2, Section 2, line 63-69. Therefore the motivation of this study could be known easily by the reviewers and readers. 

Concepts of pentamode metamaterials were proposed as early as 1995. In 2012, additively manufacture enabled the realization of such light-weighted materials. Compared with traditional materials, mechanical properties of pentamode metamaterials were linked to their macro-topology. With macro-topology adjustment, negative thermal expansion and negative Poisson´s ratio were achieved [17-19]. Thus, thermal stresses and strains in structures might be accommodated by pentamode metamaterials reinforcement.

Comment 7.  Formatting suggestion: Fig. 2: caption of (b) is missing. ---Figure 3b: could the authors mark the end of heating?  ---Fig. 6 a, b are missing; please move the legend in Fig. 6a to reveal the entire T6 curve.

Reply:  Correct

In the revised manuscript, Fig. 2, Fig. 3b, and Fig. 6 were rewritten as formatting suggestions.

Fig. 2 caption was rewritten as:

"Fig. 2 Illumination of designed heat pipes: (a) with pentamode metamaterials reinforcement, (b) without pentamode metamaterials reinforcement,".

In Fig. 3b, evaporator (heating area) of heat pipes were displayed. Thus, it was easily to known that the end of heating was 250 mm long from one endcap. 

Fig. 6 caption was revised as " Startup results of heat pipes at 850oC heating: (a) HP1, (b) HP2 ", and the legend in Fig. 6a was moved to reveal the entire T6 curve, seen below:

Fig. 3

Fig. 6

Comment 8.  Minor English language suggestion: Please check the use of articles and tense throughout the manuscript.

Reply:  Correct

English language was double-checked throughout the manuscript. The revisions were as following:

Page 1, line 20, "Heat pipe" was revised as "Heat pipes"

In Pages 1, line 28, "For, power throughputs of 15 kW/cm2 were possible for sodium heat pipes operated at 880oC [5]" was revised as:

 "Power throughputs of 15 kW/cm2 were possible for sodium heat pipes operated at 880oC [5]. "

In page 12, Conclusions, line 278-279, "In this study, pentamode metamaterials were proposed for thermal stress accommodation of high temperature heat pipes." was revised as:

"In this study, pentamode metamaterials were proposed for thermal stress accommodation of alkali-metal heat pipes."

Replies to Reviewer  #3

This article presents a novel design of heat pipe reinforced with a pentamode metamaterials infill. Thermal and mechanical properties are characterization in the application setting. The results presented support the conclusion that the pentamode metamaterial infill can alleviate the unfavoured stress distribution in the heat pipe during its operation. 

A few questions and suggestions listed as the following:

Comment 1.  The experiment design stated that HP1 and HP2 are both manufactured with pentamode metamaterial infill, but the results suggested that HP1 and HP3 are the two with infill (line 201-203). Please clarify in Table 1 and double-check throughout the manuscript.

Reply:  Correct

In the revised manuscript, HP1, HP2, HP3, and HP4 were clarified in Table 2 Heat pipes with and without pentamode metametarials reinforcement). Then the manuscript was double-check. Statement in experiment design was rewritten as: 

In Table 2, HP1 and HP3 were with pentamode metamaterials reinforcement, while HP2 and HP4 were without pentamode metamaterials reinforcement.

Table 2 Fabricated heat pipes with and without pentamode metametarials reinforcement

Models

Model

weight (g)

Charged

sodium (g)

Charging

ratio (%)

Density

(g/cm3)

Pentamode

Metamaterials

HP1

500.3

16

8

2.04

With

HP2

495.0

16

8

2.02

Without

HP3

498.2

20

10

2.03

With

HP4

497.2

20

10

2.03

Without

Comment 2.  Please specify the direction of the stress and deformation in Fig. 11 and Fig. 12, respectively. Also updated the corresponding sections in the results and discussion.

Reply:  Correct

In the revised manuscript, stress and deformation in Fig. 11 and Fig. 12 were specified. Then the corresponding section in the results and discussion was updated, seen in Page 9-10, line 214-228, Fig. 11, and Fig. 12.

Fig. 11 shows Mises stresses located in TC6 of sodium/Inconel 718 heat pipes. Mises stresses were unidirectional. At 650oC heating, Mises stresses in HP3 were 10.2 MPa. At 750oC, 850oC, and 950oC heating, Mises stresses in HP3 increased to 16.0 MPa, 29.2 MPa, and 52.4 MPa, respectively. On the other hand, Mises stresses in HP4 were 12.9-62.1 MPa, being about 30% higher than that in HP3. With stress accommodation, sodium/Inconel 718 heat pipes with pentamode metamaterials reinforcement (HP3) could be used more confidently.

Fig. 12 shows axial deformation of HP3 and HP4 at 650-950oC heating. The direction of axial deformation was from evaporator to condenser (from TC1 to TC6). In Fig. 12, we could see that pentamode metamaterials reinforcement did not affect axial deformation of sodium/Inconel 718 heat pipes. At 650oC heating, axial deformation of HP3 was 5.07 mm. As heating temperature rose, the axial deformation of HP3 increased linearly to 8.01 mm. Moreover, axial deformation of HP4 was almost the same to that of HP3. This was beneficial for engineering applications of sodium/Inconel 718 heat pipes with pentamode metamaterials reinforcement.

Fig. 11

Fig. 12

Comment 3.  What exactly does the pentamode infill do to the structure? Would the authors provide a zoom-in image of the simulation and corresponding analysis focusing on the pentamode metamaterials, in addition to the one currently only showing the entire tube?

Reply:  Correct

In the revised manuscript, a zoom-in image of the simulation was provided in Fig. 10, explaining what exactly does pentamode metamaterials infill do. Then corresponding analysis focusing on the pentamode metamaterials were added, seen below:

Analysis added in Page 9, first Paragraph, line 211.  

A further investigation indicated that this accommodation was related to pentamode metamaterials reinforcement (seen zoom-in images in Fig. 10), which would be discussed in section 3.3.2.

Analysis focusing on the pentamode metamaterials was added in Page 11, section 3.3.1, line 267.

With pentamode metamaterials reinforcement, elastic strains occurred not only in the heat pipe walls, but also in the reinforcement, seen zoom-in images in Fig. 10. Because Poisson´s ratio of pentamode metamaterials was negative, strain areas in the reinforcement were larger than that in the heat pipe walls [19, 27, 28]. Therefore, stresses in the heat pipes were accommodated.

Fig. 10 Thermal stresses (Mises) in sodium/Inconel 718 heat pipes at 850oC heating.

Comment 4.  The authors presented a relatively unclearly discussion about the effect of pentamode infill on the mechanical behavior of the heat pipe. Which properties of the pentamode metamaterials (negative Poission̕s ratio, or negative thermal expansion? Or combination of them) cause the reduction of stress?

Reply:  Correct

A new paragraph was added in Page 11, section 3.3.1, line 267, indicating that negative Poission̕s ratio cause the reduction of stress.

Effects of pentamode metamaterials reinforcement were properly as following. During startup tests, heat was transferred from evaporator to condenser. Stress concentration and elastic strains were produced in an area of heat pipe walls. And this area moved from evaporator to condenser. With pentamode metamaterials reinforcement, elastic strains occurred not only in the heat pipe walls, but also in the reinforcement, seen zoom-in images in Fig. 10. Because Poisson´s ratio of pentamode metamaterials was negative, strain areas in the reinforcement were bigger than that in the heat pipe walls [19, 27, 28]. Therefore, stresses in the heat pipes were accommodated.

Comment 5.  In section 3.3.2, the authors ran the simulation with applied external pressure. During the operation of heat pipe, the pipe should go under internal pressure. Could the authors discuss and bridge this discrepancy?

Reply:  Correct

In the revised manuscript, pressure discrepancy was discussed in section 3.3.2, Page 11, line 260-263.

According to equation (1), we could know that sodium pressure (Ps) in this study was about 4.6×104 Pa, being lower than atmosphere pressure (1.0×105 Pa). Consequently, a pressure of 5 MPa was put on the outside walls of HP3 and HP4, illuminating effects of pentamode metamaterials reinforcement.

Comment 6.  A question just for curiosity, normal the heat pipes will be manufactured through an extruding process? Would it be possible to introduce the pentamode metamaterial infill after the tube is extruded?

Reply:  Correct

In the revised manuscript, pentamode metamaterial was induced, seen in Page 2, Section 2, line 63-69. Therefore the motivation of this study could be known easily by the reviewers and readers. 

Concepts of pentamode metamaterials were proposed as early as 1995. In 2012, additively manufacture enabled the realization of such light-weighted materials. Compared with traditional materials, mechanical properties of pentamode metamaterials were linked to their macro-topology. With macro-topology adjustment, negative thermal expansion and negative Poisson´s ratio were achieved [17-19]. Thus, thermal stresses and strains in structures might be accommodated by pentamode metamaterials reinforcement.

Comment 7.  Formatting suggestion: Fig. 2: caption of (b) is missing. ---Figure 3b: could the authors mark the end of heating?  ---Fig. 6 a, b are missing; please move the legend in Fig. 6a to reveal the entire T6 curve.

Reply:  Correct

In the revised manuscript, Fig. 2, Fig. 3b, and Fig. 6 were rewritten as formatting suggestions.

Fig. 2 caption was rewritten as:

"Fig. 2 Illumination of designed heat pipes: (a) with pentamode metamaterials reinforcement, (b) without pentamode metamaterials reinforcement,".

In Fig. 3b, evaporator (heating area) of heat pipes were displayed. Thus, it was easily to known that the end of heating was 250 mm long from one endcap. 

Fig. 6 caption was revised as " Startup results of heat pipes at 850oC heating: (a) HP1, (b) HP2 ", and the legend in Fig. 6a was moved to reveal the entire T6 curve, seen below:

Fig. 3

Fig. 6

Comment 8.  Minor English language suggestion: Please check the use of articles and tense throughout the manuscript.

Reply:  Correct

English language was double-checked throughout the manuscript. The revisions were as following:

Page 1, line 20, "Heat pipe" was revised as "Heat pipes"

In Pages 1, line 28, "For, power throughputs of 15 kW/cm2 were possible for sodium heat pipes operated at 880oC [5]" was revised as:

 "Power throughputs of 15 kW/cm2 were possible for sodium heat pipes operated at 880oC [5]. "

In page 12, Conclusions, line 278-279, "In this study, pentamode metamaterials were proposed for thermal stress accommodation of high temperature heat pipes." was revised as:

"In this study, pentamode metamaterials were proposed for thermal stress accommodation of alkali-metal heat pipes."
